# Economic Impact of Lean Healthcare Implementation on the Surgical Process

**DOI:** 10.3390/healthcare12050512

**Published:** 2024-02-21

**Authors:** Marc Sales Coll, Rodolfo De Castro, Anna Ochoa de Echagüen, Vicenç Martínez Ibáñez

**Affiliations:** 1Essentia Health Management, 08021 Barcelona, Spain; aochoa@essentiahm.com; 2Department of Organization, Business Management and Product Design, University of Girona, 17003 Girona, Spain; 3Vall d’Hebron Hospital Universitari, 08035 Barcelona, Spain; vmi1@mac.com

**Keywords:** lean healthcare, hospital management, economic impact, surgical process

## Abstract

Objectives: The objective of this study was to analyse and detail surgical process improvement activities that achieve the highest economic impact. Methods: Over 4 years, a team of technicians and healthcare professionals implemented a set of Lean surgical process improvement projects at Vall d’Hebron University Hospital (VHUH), Barcelona, Spain. Methods employed in the study are common in manufacturing environments and include reducing waiting and changeover time (SMED), reducing first time through, pull, and continuous flow. Projects based on these methods now form part of the daily routine in the surgical process. The economic impact on the hospital’s surgical activity budget was analysed. Results: Process improvements have led to annual operational savings of over EUR 8.5 million. These improvements include better patient flow, better management of information between healthcare professionals, and improved logistic circuits. Conclusions: The current cultural shift towards process management in large hospitals implies shifting towards results-based healthcare, patient-perceived value (VBHC), and value-added payment. A Lean project implementation process requires long-term stability. The reason a considerable number of projects fail to complete process improvement projects is the difficulty involved in establishing the project and improving management routines. Few studies in the literature have investigated the economic impact of implementing Lean management a posteriori, and even fewer have examined actual cases. In this real case study, changes to surgical block management were initiated from stage zero. After being carefully thought through and designed, changes were carried out and subsequently analysed.

## 1. Introduction

Lean healthcare lends a new approach to the process management of surgical blocks and boosts resource efficiency by adapting the process to the high variability in patient waiting lists. The surgical process is one of the main economic outlays in hospital budgets, as well as being a main source of income [1,2]. It is widely known that managing surgical resources is critical to the financial sustainability of hospitals [3]. Improvements to the surgical process have a significant economic impact, and funds that are released can be reinvested to increase surgical capacity. A hospital with more funding is in a better position to increase both the value perceived by patients and overall health results.

In recent years, Spanish hospitals have been adopting a Lean methodology in process management. Despite the challenges posed, some have managed to consolidate an operations management model based on Lean concepts transferred to the field of healthcare [4], particularly the surgical process.

Currently, the Lean healthcare process management model and the value-based healthcare (VBHC) model are fully in place. These projects have progressed from short-term experiments in specific areas to strategic hospital projects led by management using Hoshin Kanri methodologies.

Thus, process departments have been able to establish long-term projects which consolidate management models and improvement actions that focus on process efficiency and have a major economic impact.

The primary goal of process improvement in public hospitals is not to seek economic impact or resource efficiency, but rather, to increase patient safety and to improve clinical practice and user accessibility. However, improvement actions that impact process efficiency clearly lead to operational savings.

In the present case, the economic impact of the new surgical process has improved patient communication and logistics flows and detected deviations from the expected standard, in real time. These changes were implemented by an improvement team comprising healthcare professionals at the hospital who were able to improve activity indicators successfully.

Savings on resources and reduced costs, coupled with improved health outcomes, provide us with the best basis for VBHC by increasing both the value of the hospital and its healthcare professionals.

The literature on Lean improvement projects in surgical areas includes an increasing number of real examples of project implementation, all of which demonstrate the considerable challenges of carrying out these projects [5]. However, there are still very few studies on the economic impact of improvements to the surgical process once the projects have been consolidated [6].

Lean methodology has helped improve health services (from emergency services [7] to outpatient processes [8]) and make them more efficient by identifying non-value added (NVA) activities and reducing them [9]. Both patient waiting time and the length of the stay in hospital are shortened, both of which contribute to providing an accessible and efficient service. Furthermore, with the right support, these projects can help healthcare organizations meet standards or goals related to timely, effective care. Key to improving patient flow is understanding the relationship between capacity and demand, and in this context, the Lean method offers essential support [7,10].

However, we are generally only presented with a theoretical framework for adapting Lean criteria, and its battery of management tools, to the healthcare environment. There is little or no translation of these concepts into the actual language of healthcare professionals. Thus, the aim of this study is to analyse and detail the improvement actions that have the greatest economic impact on the surgical process.

Hospital managers need to exercise good budgetary control, report to public authorities on health outcomes, and more importantly, guarantee minimum patient waiting times for surgical interventions. Thus, this study provides an established system with which to assess the economic impact of managing a surgical block based on Lean principles.

We document the basis of the economic impact of optimal surgical management and propose a series of indicators that show the evolution of results with respect to the cost-effectiveness of each euro invested and the health results obtained. Rothstein’s [11] theoretical framework for increasing operating room efficiency outlines the cost-reducing Lean tools that increase safety and patient satisfaction and motivate healthcare professionals. It also shows us optimal indicators for measuring the efficiency of the surgical process and suggests scenarios for implementing Lean projects.

Even so, few cases of the long-term economic impact of Lean management of the surgical process could be found in the literature. It is, therefore, difficult to compare the Vall d’Hebron case study with other European hospitals, despite similar experiences in Holland, Sweden [12], and England [13]. In the case of NHS England, although the management model has shown operational impact, it has been unable to generate an economic impact based on operational savings reinvested into providing care to more patients. For many trusts, this has led to situations that are no longer tenable.

The (economic) results of applying our method for more than 4 years in a prominent hospital with a large-scale surgical unit are presented in Section 3 in graphical and numerical form.

## 2. Methods

This project was carried out at Vall d’Hebron University Hospital (VHUH) in the framework of improving the hospital’s surgical process. The project became a lever for cultural change, determined the need for new ways of working, and provided an opportunity to use Lean methodology in its design and implementation.

The methodology used is within the scope of Action Research, which has already been used on other occasions to show experiences in the healthcare environment. This led us to using mixed methods that included work team participation and analysing secondary data. These particular methods are often used in Lean manufacturing environments and include reducing changeover time (SMED), reducing the time to first good part (first time through), pull, and continuous flow within a DMAIC framework.

The hospital performs over 38,000 operations annually and is a centre of reference for high-complexity surgery and organ transplants in Spain.

The management’s commitment [14] is key to the success and sustainability of Lean projects, along with the participation and training of all healthcare professionals involved. A major transformation project such as Lean management of health processes can only be sustainable and enduring if there is strong leadership, as it takes considerable effort to convince staff to implement and maintain new procedures [15,16].

The hospital used Lean Healthcare, Design Thinking, and Hoshin Kanri [17,18,19,20,21,22] methodologies to establish a patient-centred process management model. The outcomes of the improvement projects implemented were analysed using key indicators and data from the period 2015–2018 and through implementing continuous improvement tools.

After 4 years of implementing the projects [23], an established work system helps us identify the key indicators of the process and a formula for improving the process and shows how the improvements have had a positive economic impact, enabling management to reinvest in the process.

Framework of indicators that affect the economic management of the surgical process.

The surgical process implemented considers operational aspects of managing patient flow by adapting Lean methodology improvement principles. These include adapting resources to demand, detecting the value provided in each process, highlighting value flow, and continuous process improvement. This study does not consider the variable costs of surgery materials for specific procedures, nor the number of human resources invested in the process, as each hospital has a different surgical care model and economic management depending on the type of hospital (public, private, or university).

The economic impact of implementing a Lean surgical process improvement project can only be assessed when the different phases of patient flow are analysed over the long term, as a sufficiently long period of time is needed to consolidate the improvements surgical staff suggest.

The actions that need to be taken are based on improving patient circuits, internal communication and planning the logistics channels that affect the surgical process.

In improvement projects, the more progress made, the greater the economic impact. The indicators used for the Lean improvements proposed for the process are as follows:Evolution of Surgical Activity (increased healthcare activity while maintaining fixed structural costs and stable human resources). Proposal → Define actions to increase activity based on making the best use of operating room hours. Increase occupancy and surgical rotation.Evolution of the Surgical Waiting List (SWL) (shortened patient waiting time for surgery, without a budget increase so more surgical space can be provided). Proposal → Analyse the relationship between increased surgical activity and the combinations of surgeries that favour shortening the SWL more quickly and ensure that patients are added to the list in a way that facilitates shorter waiting times.Surgical Occupation (time with the patient inside the operating room in relation to the standard work shift). Proposal → Knowing that operating room costs per minute are high [2], the optimal surgical employment ratios per service must be defined so professional healthcare teams reach maximum efficiency, with maximum rotation of patients per surgical slot.First Case On-Time Starts (FCOTS) (all operating rooms must start the first patient’s “skin opening” surgery before 8:30 a.m.) [24,25]. Proposal → Speed up operating room preparation and patient arrival processes to facilitate correct, prompt start of surgery.Average Pre-Operative Stay (time patient spends in hospital before undergoing surgery). Proposal → Design the best clinical pathways for carrying out the surgical program without having to lengthen the patient’s pre-surgery stay for reasons unrelated to healthcare.Reprogramming Index (patient changes in confirmed surgical schedule). Proposal → Eliminate or minimize changes to the surgical schedule in the 24 h prior to, or on the same day as the scheduled surgery to avoid loss of surgical resources.Quick Patient Changeover (operating room turnover time, or TOT). Proposal → Streamline and speed up patient changeover processes between interventions in order to optimize operating room occupancy times and activities carried out by the healthcare teams. Improve operating room preparation of surgical instruments required for the next patient [26].Average Hospital Stay (time patient spends in hospital post-surgery). Proposal → Design the best clinical pathways for post-surgical recovery without increasing a patient’s post-surgery stay for reasons unrelated to healthcare.

These baseline indicators provide ground for determining each block’s potential for improvement and establish a roadmap for implementing process actions that increase the efficiency of hospital resources.

## 3. Results

The initial situation encountered at the VH hospital in 2015 showed of a lack of effective management in most of the above indicators. However, the very different situation in 2018 shows the remarkable effort made by the surgical teams in the new surgical block to improve circuits and the management of the clinical trajectory of each patient, from their addition to the SWL to their post-surgery stay in the hospital and eventual discharge.

Figure 1 shows the baseline assessment of the project towards the end of 2018 and the results of the improvement projects implemented.

Economic Impact of Improvement Projects

Four years after initiating the surgical process improvement projects, the hospital’s finance department has made the economic data available, thus enabling the mesurement of the economic impact of changes in VH Hospital’s process management, based on the good results of the indicators and the reliability of the data.

The main aim of a large hospital is not to save money on the budget, but rather pursue two larger goals:(1)Operate the planned annual budget without creating a deficit.(2)Maximize healthcare for the maximum number of patients within the allocated budget.

It is worth noting that all savings in the process should be understood as operating savings that are reinvested in the block, so more patients can access the surgical care they need.

In the VH Hospital’s 60-year history, the zero deficit target been only ever been reached twice: in 2017 and again in 2018. These same years also yielded the best results for waiting list management and overall patient satisfaction.

To analyze the impact of each of the above indicators on improving the economic performance of the surgical process, we need to distinguish between increasing income and increasing activities that generate a higher turnover and reduce costs. These activities include the following:managing medical equipment;optimising length of stay in pre- and post-surgical wards according to the needs of each clinical pathway, as well as improving the discharge process [27];reducing the number of additional tests with an improved preoperative process which includes a reorganised scheduling system and anesthesia management agenda;avoiding rescheduling operations as this leaves gaps in the operating room agenda, and surgical equipment sits idle;managing surgeons’ agendas for first diagnostic and consecutive pre-operative visits with little or no care value.

These factors have contributed to average annual savings of EUR 8.5 million on a turnover of EUR 115 million, which is the hospital’s budget for surgery (Figure 2. Surgical area billing) (Reprinted/Adapted with permission from Vall d’Hebron University Hospital, 2018 [28]). Moreover, there is an increase in annual profit due to the higher number of patients operated on.

In this case, after reorganising surgical processes, annual savings amounted to 7.4% of the total cost of a high-complexity surgical block, such as the one in VHUH (categorised in Figure 2). This demonstrates that the hospital was able to increase surgical activity using the same number of operating rooms, as the structural cost of professional healthcare teams and maintaining physical resource (e.g., operating rooms) are already included in the planned budget. However, variable costs stemming from consumable material [29], such as prostheses, implants, additional instrument boxes [26], etc., have been covered by savings in other areas and the Department of Health’s achievement of its billing commitments from the increased activity generated.

In the items grouped under operating savings, almost EUR 5 million is derived from reducing pre- and post-surgery hospital stays. The hours of healthcare saved on hospital wards pre- and post-surgery has been reinvested in caring for patients in the emergency department, or attending to new patients on the SWL as more hospital beds are made available.

Over the course of 2018, the increase in surgical activity resulting from the highly efficient use of operating theatre resources meant that the programme was overloaded. However, this was due to a shortage of in-patient beds rather than a lack of surgical resources, prompting some rethinking of the improvement projects. A new focus centred on hospital drainage, rapid discharge procedures, the clinical pathway established, and, most importantly, launching Short Surgical Stay Units [30] that could keep pace with surgical programming and the daily scheduling of operations, which was subject to the rate of hospital discharges. A specific in-patient ward was also opened on Monday at midday and closed again on Saturday once all surgical patients had been discharged from the hospital.

This key operational savings project increased activity without increasing costs, and the real economic savings lie in obtaining a process that does not require care staff 48 h a week, so efficiency and employment indicators remain unaffected.

The remaining operating savings are:Reducing complementary testing: clinical practice analysis contrasted with complementary tests by hospital healthcare professionals for each surgical procedure is required for decision-making. We detected an overuse of diagnostic testing.Reducing the need to reschedule: from standardizing surgical scheduling and keeping spaces available in agendas and flexible operating rooms for deferred emergencies, it was practically impossible to prevent any operating room from losing surgical activity due to 24 h rescheduling and surgical use, which was close to 98%.Optimising management of surgical equipment: by analysing the intermediate inventories of the warehouse chain, the material kits for each surgery could be prepared and material allocated to the patient in advance, thus optimising the management of the logistics circuit. To do this, VSC (Value Stream Costing) was used. This is a process of identifying and establishing costs for all the steps of the logistics process that are necessary to provide value to the system. This function determines how much value, or cost, is created in each part of the process, and tracks all the steps associated with the activity. Value flow mapping is a key element of Lean thinking. It focuses on providing value to the customer and identifying waste in the process. In parallel, surgical cost savings can be achieved by having lists of materials needed for standard procedures and paying attention to the cost of consumable surgical supplies [29].Reducing first surgical and pre-surgery anaesthesia visits: with increased activity, it became apparent that there was a lack of space for pre-surgery consultations and anaesthesia services lacked the capacity to perform pre-surgery tests. Thus, the number of pre-surgery visits was reduced to those that were strictly necessary, with some of the face-to-face appointments with nursing staff changed to online visits. The quality of care improved, and the growing demand was met by adapting the human and material resources available.

The measures taken during the 4-year project have saved the hospital EUR 25.5 million in operation costs over the last 3 years. As surgical process indicators improved; savings increased.

From 2018 onwards, the model was consolidated, and the indicators were maintained by applying Hoshin Kanri methods in the follow-up strategy. The economic impact of the changes has already been integrated into process routines, and ongoing improvement actions are able to spotlight the parts of the process where it is still possible to obtain better efficiency ratios.

## 4. Discussion

The new Vall d’Hebron University Hospital Surgical Block is an exceptional testing ground for this unique improvement project. The opportunity to transform from an outdated, segregated system of operating rooms to a new joint facility with high technical performance has enabled the cultural change required to modify old routines and redesign circuits and processes. We are leading the way towards Lean management within healthcare in Europe, while always focusing on patient safety, patient flow, resource efficiency, and communication between professionals, which all constitute areas of improvement.

Results for 2018 show a continuous increase in surgical activity stemming from the various surgical efficiency projects, such as a continuous, uninterrupted operating room schedule (morning + afternoon) [4] for all surgical services.

The hospital has increased its outpatient surgical activity by 22% and conventional surgery by 7%; however, demand for surgery has also grown. Between 2014 and 2018 the number of patients on the SWL increased by 23.5%. This demonstrates the high capacity and efficiency of resources at the hospital, which has been able to absorb the surge in activity without increasing the number of operating rooms.

Considering that the cost of an operating room is EUR 1200/hour, surgical performance and activity have increased without the need for more operating room hours. In this case, surgical performance increased, so there was no need to request the opening of more operating theatres to cope with the increased surgical demand of the population.

The Preoperative Mean Stay (PMS) is another important indicator that improved under the new management system. In the past, patients were admitted the day before surgery in order to ensure that they were present and fit to be operated on, and staff could prepare them for the operation. In contrast, the current process admits the patient on the day of the operation, which benefits both the patient and the hospital. On one hand, the patient can sleep at home the night beforehand, where they are accompanied and rest better, and can arrive at the hospital 2–3 h before the operation. On the other hand, the hospital saves beds and can allocate nursing resources and empty beds to emergency rooms or other patients who require them.

A pre-admission area was created to ensure good patient flow. This is where the preparation and transfer processes begin, and from 6:30 am, specialized nurses welcome patients and guide them towards the surgical block. This new pre-admission area has reduced the average length of hospital stays by 78%. It ensures correct admission, improves patient preparation and comfort, and guarantees the expected first round of surgical patients. This improves FCOTS exponentially and has a direct impact on the use of surgical resources [24].

In 2018, the PMS index for all surgical patients decreased to 0.3 days (7 h), compared to 1.4 days (33 h) in 2014. In general, the length of stay in hospital before surgery has been cut by 1 day per patient. The Department of Health sets the cost of hospitalisation at EUR 400/night and the savings achieved annually at VHUH are around EUR 550,000/year, considering that they operate on approximately 19,000 scheduled patients annually, 1375 of whom are first surgical interventions. However, data worth highlighting are the marked reduction in post-surgery hospital stays, by virtue of defined clinical pathways that include an established discharge process. This means 11,000 fewer overnight stays for over 7300 patients, and annual savings of EUR 4.4 million. These operational savings are reinvested in caring for more patients, while using the same resources, and increasing overall hospital discharges.

Each of these indicators has a positive impact on the financial management of the surgical block, although this may go unnoticed within the framework of the large annual budget (more than EUR 750 million per year) managed by the finance department of a large hospital such as VHUH. An operational saving of this nature, however, can make the difference between overspending or not.

## 5. Conclusions

While implementing the surgical process improvement projects, we identified several indicators for the evolution of cultural change among the healthcare professionals towards results-based processes [31].

A new surgical management system was implemented, providing transparency, reliability of data collected, and a vision of new opportunities for improved patient flows and the flow of internal communication and logistics.

Implementing this Lean project was crucial to improving results [32]. However, it necessarily required a first phase of designing circuits, structuring, and detecting the value of each stage of the process, reducing variability and establishing valid management indicators. This was followed by the implementation of the improvement project, followed by a monitoring stage which consolidates the work completed and optimises resources in the hospital’s surgical block [33].

The final project provides five major blocks of knowledge (Figure 3) and tools for management that follow an Integral Surgical Management System structure:Real-time evolution of the surgical processStandardised programming with pre-established efficiency indicatorsMedium- and long-term planning in order to adapt available resources to demandPlanned economic management of the surgical block by standardising the necessary time and materials requiredImproved patient experience and information shared with family members throughout the process

To conclude, we must emphasise that a healthcare management model such as this must prioritise the surgical patient’s safety and satisfaction with the process and improve clinical practice so it can adapt to new techniques based on a culture of measuring the value given to the patient.

Savings from an efficient surgical process must be reinvested in the process in order to improve user accessibility within the maximum guaranteed time for surgery and must also improve patient or professional satisfaction due to maximum coordination of surgical resources for the project to have the desired effect and a patient-centred model of care.

## Figures and Tables

**Figure 1 healthcare-12-00512-f001:**
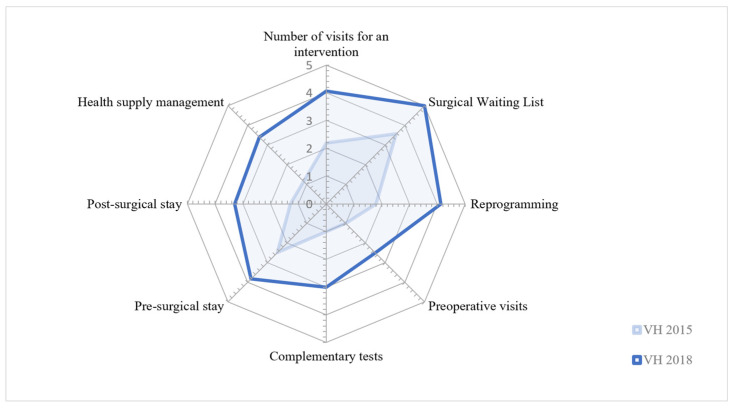
Evolution of improvement areas generating economic impact.

**Figure 2 healthcare-12-00512-f002:**
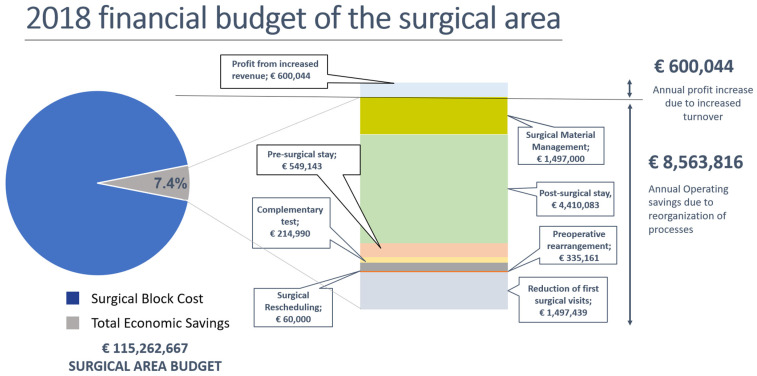
Surgical area billing.

**Figure 3 healthcare-12-00512-f003:**
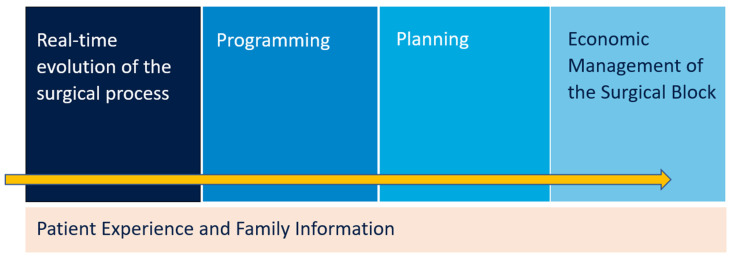
Five major blocks of knowledge and tools for management involved in improvement process.

## Data Availability

Data are contained within the article.

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
