# Peer review of "Economic Impact of Lean Healthcare Implementation on the Surgical Process"

_healthcare, 2024, doi:10.3390/healthcare12050512_

Round 1

Reviewer 1 Report (Previous Reviewer 2)

Comments and Suggestions for Authors

This paper focuses on analyzing the economic impact of implementing lean surgical process improvement projects at Vall d’Hebron University Hospital. However, I don’t believe that the manuscript has been significantly improved with compared to the previous version.

1. The research methodology is not clearly presented in the abstract.

2.  In the section on methods, although the authors claim to have used a mixed methodology of methods, it is still considered appropriate to specify which analytical methods were used. A research paper that lacks in-depth analysis of the raw data is not convincing and lacks value in its conclusions.

3. Figures 1 and 2 lack titles.

4. How the data in Figure 2 was calculated should be described in detail. Does the data in the graph refer to 2018 or 2015–2018? The keyword "economic evolution 2015-2018" in this figure is confusing.

5. There is a lack of sufficiently detailed data in the case studies.

Comments on the Quality of English Language

Moderate editing of English language required.

Author Response

We agree that few changes have been made to the manuscript. We understand we need to be more precise when answering the reviewers’ comments and make more changes to the manuscript.

  1. The research methodology is not clearly presented in the abstract.

The research methodology has been clarified in the abstract as requested. However, we think the abstract now contains too much technical information.

  1. In the section on methods, although the authors claim to have used a mixed methodology of methods, it is still considered appropriate to specify which analytical methods were used. A research paper that lacks in-depth analysis of the raw data is not convincing and lacks value in its conclusions.

We agree with this comment and have expanded the paragraph on methods added in the previous revision. This is the same information we propose adding to the abstract in line with Reviewer 1’s previous comment.

  1. Figures 1 and 2 lack titles.

This is true. We think this was due to problems formatting the Word file. Titles have now been added to Figures 1, 2 and 3.

  1. How the data in Figure 2 was calculated should be described in detail. Does the data in the graph refer to 2018 or 2015–2018? The keyword "economic evolution 2015-2018" in this figure is confusing.

We agree with this comment and have added two sentences to clarify Figure 2. We have also changed the title of Figure 2 to give a better understanding of the numbers represented. Figure 2 represents the 2018 budget, and these savings have been matched every year as a result of the projects carried out over the three years (2015 – 2018). We wanted to highlight the increased revenue resulting from increased efficiency and the higher number of patients attended.

  1. There is a lack of sufficiently detailed data in the case studies.

We partially agree with this comment. We present the methodology and economic results based on the research action project conducted. Other related papers by the same authors can be found in the bibliography with more detailed data on our case studies.

Reviewer 2 Report (New Reviewer)

Comments and Suggestions for Authors

Dear Authors,

For sure this topic is valuable, however:

- in the introduction if (line 92) it is stated that :"existence of similar experiences in Holland, Sweden and England", which means that these countries (or in these countries) have such experience therefore, you should to make the review of this results; 

- you should work on consistency of this article as in the introduction you have mentioned about figure 2, while this figure is second and presented and described later in the text. 

- in the methodology part it is not clear - for what period the data is collected and finally the range apart from information of yearly number of operations.

- discussion should include also the discussion with the missed review of such research made for in Holland, Sweden and England"

Author Response

Many thanks for this comment. We are working to improve the manuscript to be published in healthcare.

- in the introduction if (line 92) it is stated that :"existence of similar experiences in Holland, Sweden and England", which means that these countries (or in these countries) have such experience therefore, you should to make the review of this results; 

We agree with this comment. We have expanded this paragraph (added in the previous resubmission), adding two references related to it.

- you should work on consistency of this article as in the introduction you have mentioned about figure 2, while this figure is second and presented and described later in the text. 

We agree with this comment and have removed Figure 2 because of the lack of consistency, as reviewer 2 suggests.  In the introduction we reference the Results section, highlighting that the economic results can be found in Section 3.

- in the methodology part it is not clear - for what period the data is collected and finally the range apart from information of yearly number of operations.

We agree with this comment. We have added a new paragraph in the Methods section, specifying the data collection period. Following the lean tool of continuous improvement we used, we collected data throughout the whole period. However, we compared indicators from 2015 and 2018 in order to show the findings.

- discussion should include also the discussion with the missed review of such research made for in Holland, Sweden and England”.

We agree with this comment and have added a paragraph in the Discussion section regarding experiences in those countries.

Round 2

Reviewer 1 Report (Previous Reviewer 2)

Comments and Suggestions for Authors

The authors have almost completed the revision of the paper.

Comments on the Quality of English Language

Moderate editing of English language required.

Reviewer 2 Report (New Reviewer)

Comments and Suggestions for Authors

All comments have been taken into account. It improved the clarity of this paper. 

This manuscript is a resubmission of an earlier submission. The following is a list of the peer review reports and author responses from that submission.

Round 1

Reviewer 1 Report

Comments and Suggestions for Authors

This is a an interesting and nicely written paper. I would think the paper would benefit from improvements along the following lines before it gets published:

- A more comprehensive literature would need to set the stage, including experiences in other countries and how implementing changes have reduced costs and by how much.

- The paper should tell us the details of how cost savings have been calculated. We would also need to know what the counterfactual is (cost savings compared to what).

- The paper would need some language editing, there seems to be some typos in the text.

Comments on the Quality of English Language

The paper would need some language editing, there seems to be some typos in the text

Reviewer 2 Report

Comments and Suggestions for Authors

This paper focuses on analyzing the economic impact of implementing lean surgical process improvement projects at Vall d’Hebron University Hospital. This issue is helpful to improve the management of hospitals.

 The thesis lacks a quantitative analysis of the economic impact of lean healthcare and does not provide an in-depth analysis of the core issues.
2. The paper lacks innovation and contribution in terms of research content, methodology, and findings.
3. The headlines of Figures 1 and 2 should be added (lines 169 and 202).
4. The research methodology should be introduced in detail.
5. There is a repetition of presenting both results and conclusions in the abstract, and it is recommended that only the core conclusions be presented.
6. It is recommended to select cases where detailed data and information can be obtained and to use case study methodology for the study.

Comments on the Quality of English Language

Moderate editing of English language required.